# LOOK BACK TO MOVE FORWARD: DELAY-AWARE INSTANCE SELECTION FOR ONLINE CONTINUAL LEARNING

## ABSTRACT

Supervised continual learning (CL) typically assumes that labels are available immediately after each input arrives. This is unrealistic in many streaming applications, where annotation latency is the norm. When labels arrive late, supervision for past tasks can spill into later tasks, entangling training signals and degrading current performance. We study this delayed-label setting and analyze how different delay regimes impact online CL. We then introduce a delay-aware instance selection strategy that prioritizes which late-labeled examples to use for updates based on a simple, model-utility criterion. By selecting only the most beneficial delayed instances, our approach accelerates performance recovery after task shifts and reduces the training budget when labels from multiple past tasks arrive simultaneously. Our contributions are: (i) a clear problem formulation and evaluation protocol for online continual learning with delayed labels; (ii) an empirical analysis across delay regimes showing how label latency mixes supervision across tasks; and (iii) a delay-aware instance-selection method compatible with replay-based CL. Experiments indicate consistent improvements in current-task accuracy and stability, with fewer update steps than delay-agnostic baselines.

## 1 INTRODUCTION

Supervised Learning is an Artificial Intelligence paradigm that presented remarkable results for last decades due to the ability of modeling data by mapping features and labels $f : X \to Y$ (Bishop & Nasrabadi, 2006; Mitchell, 1997). However, in many real-world deployments, systems operate on non-stationary data streams in which the underlying distribution drifts over time (Gama et al., 2014). In these settings, models must update continually to remain accurate on the current distribution, a challenge studied in Online Continual Learning (OCL) (Delange et al., 2021).

Although the OCL literature has advanced considerably, introducing methods that learn from unbounded data streams and progressively extend acquired knowledge while mitigating catastrophic forgetting (McCloskey & Cohen, 1989), most advances implicitly assume that labels are immediately available. This assumption is unrealistic in many real-world pipelines, where annotation is delayed by human labeling cycles, batched ingestion, privacy reviews, or edge-to-cloud transfer. When labels arrive late, updates for earlier inputs spill into later phases of the stream, interfering with learning on the current task.

Concretely, inputs $x_t$ are observed at time $t$, but their labels $y_t$ become available at $t + \Delta_t$, with $\Delta_t > 0$ drawn from a delay regime. Under distribution shifts (task boundaries), late supervision from earlier tasks spills into later tasks, entangling training signals across tasks and degrading current-task performance and stability. Figure 1 illustrates this effect: relative to immediate labels, delayed labels depress accuracy and slow recovery after a shift.

Our empirical study shows that the nature of the delay regime matters. With small delays, supervision drift is mild but non-negligible; with moderate delays, the training signal substantially overlaps multiple tasks; with large or bursty delays, the learner faces update floods where labels from several past tasks arrive simultaneously. In all cases, blindly consuming every delayed label wastes updates and can bias the model away from the current distribution.

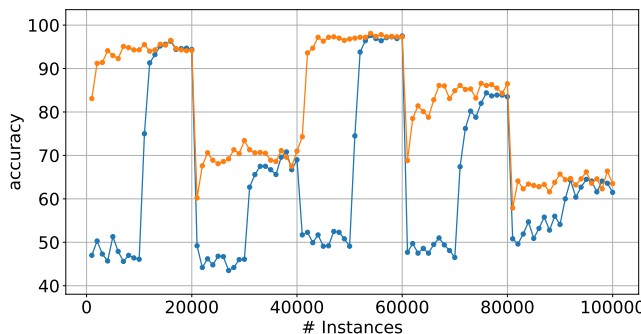

Figure 1: Effect of label latency on online CL. Orange (—●—): accuracy per task with immediate labels. Blue (—●—): accuracy when labels arrive late, causing updates for past inputs to occur in later tasks and depressing current-task performance.

To address this, we introduce a plug-and-play delay-aware instance selection strategy. Rather than updating on all delayed labels, we estimate a model-utility criterion for each labeled past instance and select only those expected to maximize current progress. This prioritization accelerates recovery after task shifts and reduces the training budget, especially under bursty arrivals. Our method is architecture-agnostic and compatible with standard replay-based CL pipelines.

Our contributions are threefold: (i) a clear problem formulation and evaluation protocol for online continual learning with delayed labels; (ii) an empirical analysis across delay regimes that quantifies how label latency mixes supervision and impacts stability; and (iii) a delay-aware instance-selection method that can be dropped into replay-based CL without modifying model architectures.

Experiments indicate consistent improvements in current-task accuracy and stability with fewer update steps than delay-agnostic baselines, demonstrating that carefully choosing which delayed instances to learn from is both effective and efficient. We hope this setting and baseline will help the community systematically study label latency in OCL and develop delay-robust learners.

## 2 PROBLEM DEFINITION

In OCL, the learner is confronted with a potentially never-ending stream of data. At every time step, a sequence of labeled examples $(x_t, y_t)$, with $x_t \in X$ and $y_t \in Y$, is produced from a distribution $D_t$, which may shift to $D_{t+1}$ at a task switch, requiring the learner to adapt to new data while mitigating forgetting of previously acquired knowledge.

Following the approach presented by Caccia et al. (2021), the learner is neither explicitly informed when a task switch occurs nor provided with a task identifier during training. On the other hand, in the OCL setting, a model receives a single pass over data stream and must update on the fly under limited memory and compute. We focus on the Online Task-Incremental setting, in which new tasks arrive over time with disjoint class sets and evaluation is performed per task, i.e., The task identity is accessible exclusively at test time and is used for evaluation purposes (Mai et al., 2022). In our setting, predictions are made across the classes of all tasks. This more challenging scenario is often termed the single-headed setup, where a single output head covers all task classes (Van de Ven et al., 2020; Farquhar & Gal, 2018).

In the recent OCL literature, most algorithms are designed for streams where the label becomes available immediately after prediction, that is, the learning process assumes no delay in label availability (Davalas et al., 2024; Van de Ven et al., 2022; Soutif-Cormerais et al., 2023; Bidaki et al., 2025). This assumption often fails in real-world systems, since labels may be delayed or missing due to transmission failures, data acquisition errors, or limited access to qualified annotators.

The relationship between the presence of labels and their availability can be formally expressed as a temporal-mapping function $T(\cdot)$ that precisely extracts discrete time unit $t$ when $x_t$ and $y_t$ are

available. Based on this function, Gomes et al. (2022) identifies four distinct definitions that describe the relationships between data and label availability in streaming data scenarios:

1. **Immediate and fully labeled:** $\forall x_t \in X,\ \forall y_t \in Y,\ \Delta_t = T(y_t) - T(x_t) = 1$. The latency validation between $x_t$ and $y_t$ corresponds exactly to a one-time unit.

2. **Delayed and fully labeled:** $\forall x_t \in X,\ \forall y_t \in Y,\ \Delta_t = T(y_t) - T(x_t) = R$, where $R$ is a random variable that represents the discrete delay between $x_t$ and $y_t$. The discrete delay is limited by the finite range $R \in \mathbb{Z}_+$.

3. **Immediate and partially labeled:** Similarly to 1, the latency validation is a one-time unit. However, only partially $X$ has a corresponding entry on $Y$. In that case, the sequence of labeled examples is $\exists x_t \in X,\ \nexists y_t \in Y,\ (x_t, ?),\ \Delta_{x_t} = T(y_t) - T(x_t) = \infty$.

4. **Delayed and partially labeled:** Based on 2 and 3, some labels are delayed, and others are missing (infinitely delayed).

In this work, we focus on *Scenario 2*, and we model the availability of current and delayed instances as mini-batches. Let the stream be partitioned into a sequence of tasks $\mathcal{T} = \{\tau_1, \tau_2, \ldots\}$, arriving sequentially over time. Each task $\tau_i \in \mathcal{T}$ contains instances whose labels belong to a subset $C_i \subseteq C$, where $C$ is the global label set and $|C_i| = m_i$ denotes the number of classes in the $i$-th task. Let $\mathcal{B} = (\mathcal{B}_1, \mathcal{B}_2, \ldots)$ denote the sequence of mini-batches. During the time interval of task $\tau_i$, the learner receives the contiguous subsequence

$$\mathcal{B}_{i:i+L_i-1} = (\mathcal{B}_i, \mathcal{B}_{i+1}, \ldots, \mathcal{B}_{i+L_i-1}),$$

where $L_i \in \mathbb{N}$ is the number of mini-batches assigned to task $\tau_i$. Each mini-batch $\mathcal{B}_b$ contains $n_b$ labeled instances and is written as

$$\mathcal{B}_b = \left\{(x_{bj}, y_{bj})\right\}_{j=1}^{n_b}, \quad x_{bj} \in X,\ y_{bj} \in Y.$$

Equivalently, with input and label collections

$$X^{(b)} = \{x_{bj}\}_{j=1}^{n_b}, \qquad Y^{(b)} = \{y_{bj}\}_{j=1}^{n_b}, \quad \text{we write} \quad \mathcal{B}_b = \left(X^{(b)}, Y^{(b)}\right).$$

In our experimental setup, an OCL method may use one or more subsets of mini-batches to update the model. To reflect limited resources, training consumes a single mini-batch at a time. When multiple mini-batches are chosen, the updates are applied sequentially.

## 3 RELATED WORK

In OCL, label delay is a significant, yet often overlooked, challenge. In this direction, Csaba et al. (2024) propose Importance Weighted Memory Sampling (IWMS), which samples from the memory buffer so that the selected set matches the distribution of the newest unlabeled data. The process has two stages: first, at each time step the model makes predictions on unlabeled samples and selects labeled samples from memory whose true labels match these predictions; second, it computes feature similarity between each unlabeled sample and the selected memory samples using cosine similarity on the learned representations. From this pool, the method samples the most relevant labeled examples according to their similarity scores. In this way, training rehearses memory samples that share the predicted labels of the unlabeled inputs and exhibit high feature similarity to them. Under the setting in Section 2, this approach is not applicable. Selecting instances from memory to infer labels for unlabeled data presupposes that batches from the current task are already available with labels. In contrast, our setting is Online Task-Incremental with delayed labels, tasks arrive sequentially without overlap, and labels may become available only after their corresponding inputs.

One of the central challenges in OCL is updating on the current stream without catastrophically forgetting previous knowledge. A simple and effective solution is Experience Replay (ER) (Chaudhry et al., 2019; Rolnick et al., 2019; Bellitto et al., 2024; **?**). ER is a rehearsal-based strategy commonly used in continual learning, which maintains a fixed-size memory buffer $M$ that stores a subset of samples encountered during training. At each learning step, the model is trained on the incoming data stream also on a batch of previously stored samples drawn from $M$ (Caccia et al., 2021). When new data arrives, it is added to the buffer, and if the buffer is already full, a replacement policy e.g.,

reservoir sampling, ring buffer, or random replacement determines which stored sample will be discarded. This mechanism ensures that the memory contains a representative set of past experiences despite the memory size being limited. By interleaving replayed samples from the buffer with the current data, ER mitigates catastrophic forgetting, allowing the model to retain knowledge of earlier tasks while still adapting to new ones. However, the algorithm's effectiveness strongly depends on the memory size, the replacement strategy, and the balance between new and replayed samples during training.

Experience Replay has also become a standard component in Reinforcement Learning, which improves both sample efficiency and training stability (Rolnick et al., 2019; Mnih et al., 2015; Fedus et al., 2020; Zuffer et al., 2025; Bellitto et al., 2024; Wang et al., 2025; Urettini & Carta, 2025; Nori et al., 2025), and has been increasingly applied in continual learning. Early approaches concentrated on the controlled sampling of stored memories for replay (Aljundi et al., 2019), reducing interference between tasks, such as using asymmetric cross-entropy to limit representation overlap (Caccia et al., 2021), and combining ER with knowledge distillation from past tasks (Buzzega et al., 2020). More recent methods enhance memory management and knowledge transfer, including dual-memory systems that align decision boundaries with semantic memories (Arani et al., 2022), and strategies that enforce prediction consistency, allowing the current model to mimic future experiences while the previous model distills past knowledge (Zhuo et al., 2023). While these studies have advanced the application of experience replay, they fail to address the OCL setting with delayed labels, which is the focus of our proposed approach.

OCL methods typically assume an unlimited computational budget for training on incoming data streams. This assumption has been increasingly contested, with recent work evaluating continual learning in scenarios where the data stream advances uninterrupted, offering no pause for model training before new samples require prediction (Csaba et al., 2024; Alfarra et al., 2025; Wang et al., 2024). This setup offers a more realistic evaluation of OCL under label-delay scenarios, where the data arrival rate may exceed the model's training capacity. Their results show that, under these conditions, all evaluated OCL algorithms perform worse than the simple ER baseline.

In this work, based on the results of Csaba et al. (2024), we examine the ER method with appropriate adaptations as a baseline, comparing it to our proposed approach for the Online Task-Incremental setting with label delays and constrained computational resources.

## 4 PROPOSED APPROACH: EXPERIENCE-DELAYED REPLAY (EDR)

In a recent large-scale study, Ghunaim et al. (2023) showed that a simple ER baseline often outperforms more complex OCL methods. Based on this finding, we examine how ER-style learners behave under label-delay regimes. Figure 2 illustrates the setup: the stream is partitioned into tasks Task 0, Task 1, Task 2, ..., and colored rectangles denote mini-batches whose labels become available within the corresponding task. Shaded rectangles indicate batches whose labels are delayed. In Task 0, for example, three green batches are labeled immediately, while a fourth (shaded green) is revealed during Task 1, as indicated by the black arrow.

In Task 1, one delayed mini-batch from Task 0 (green) arrives together with three mini-batches from the current task (blue). Under classical ER with a limited update budget, as is typical in practice, the learner consumes the earliest available batches, i.e., the delayed green batch plus the first blue batches. As a result, a non-trivial share of the update budget that should target the current task is spent on past-task data. Ideally, most updates in Task 1 should be allocated to current-task examples, while the previous task is represented only through reservoir samples (gray), rather than through late, full mini-batches. In Task 2, the issue is amplified, i.e., early training on the red task is flooded by delayed mini-batches from Tasks 0 (green) and 1 (blue). Consequently, the model under ER spends a substantial portion of its initial update budget on past-task supervision, and current-task accuracy improves only later, once ER finally processes Task-2 examples.

To mitigate this issue, we make a minimal change to ER: Random Experience Replay (RER). Instead of consuming labeled mini-batches strictly in arrival order, RER forms each update batch by sampling uniformly at random from all labeled examples available at the current time. This breaks the front-loading of delayed past-task labels and increases the share of current-task updates. In Task 1, for instance, with one delayed green batch (Task 0) and three blue batches (current task), RER,

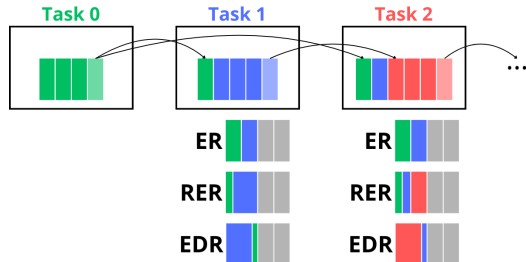

Figure 2: Data streams are represented by a sequence of tasks composed of mini-batches. To illustrate, colored rectangles show mini-batches whose labels are available within the current task, while shaded rectangles indicate delayed labels revealed in later tasks (black arrows). Gray rectangles are data from the reservoir used by ER methods. The execution of the ER, RER, and EDR methods are represented by the way how instances are selected from current and delayed mini-batches.

a

by expectation, allocates more updates to Task 1 than to Task 0, better aligning the training budget with the current distribution.

Unfortunately, this benefit holds only when the available labeled pool is dominated by current-task data. As delays accumulate (see Task 2), the backlog of past-task labels grows, and under uniform sampling the chance of selecting current-task instances drops proportionally. The result is, again, early updates skewed toward outdated supervision.

We address this with Delayed Experience Replay (EDR), a variant of ER that down-weights delayed labels. At each update, examples are sampled with probabilities that decrease with their delay, so late mini-batches from past tasks are less likely to dominate early training. Prior exposure to earlier tasks is maintained via the reservoir. However, delay alone should not determine selection: EDR also accounts for task performance/utility. Under a test-then-train strategy, for instance, past-task instances with high current loss (or low confidence) remain eligible to improve the model. Formally, $p(x_t) \propto l(x_t)\, w(\Delta_{x_t})$, where $l(x_t)$ encodes utility (e.g., loss-, margin-, or gradient-based scoring) and $w(\Delta_t)$ is a decreasing function of delay for the instance $x_t$. As shown in Tasks 1 and 2, current-task mini-batches receive higher priority because they exhibit smaller delays and larger losses while the model is still adapting to the current distribution. In Task 2, instances from Task 1 are also preferentially sampled: they have shorter delays than those from Task 0 and often higher utility, since many Task-0 examples were already consumed in Task 1 and are represented in the reservoir, reducing their novelty and loss.

A simple way to quantify the importance of delayed versus current instances is given by Eq. 1, which weights the loss by a delay-dependent factor. In this scheme, misclassified instances have their contribution attenuated by a factor $k > 0$ according to the delay magnitude.

$$I\big((x_t, y_t), \Delta_t\big) = \left(-\sum_{c=1}^{|C|} y_{t,c} \log \hat{y}_{t,c}\right) \cdot e^{-k \cdot \Delta_t} \tag{1}$$

## 5 EXPERIMENTAL METHODOLOGY

We evaluate the impact of label delays in OCL using an experimental setup implemented in Capy-MOA (Gomes et al., 2025), a machine-learning library for data streams. To simulate incremental learning, classes are partitioned into sequential tasks. Performance is measured with a test-then-train protocol, such that each incoming instance is first evaluated and then used for updating. In the Online Task-Incremental setting, the task identity is provided at test time, and predictions are restricted to the corresponding task's label space. The evaluation stream may contain instances from current and past tasks, and, as previously mentioned, we do not assume unlimited computational budget. Consequently, delayed instances arrive concurrently with instances from the current task. In our experiments, we assess how different approaches handle all avail-

able instances under this overlap. Code and datasets used in this experiments are available at https://anonymous.4open.science/r/CapyMOA-4AD7.

**Learning Model.** We assess how learning reacts to label delays on computer-vision tasks using deep neural networks based on published architectures. Our aim is not to improve these backbones, but to evaluate their behavior under delay. Accordingly, we adopt standard implementations and training protocols from the literature and hold architectures fixed across OCL methods. In summary, the adopted architecture was based on a two-layer MLP with a ReLU nonlinearity between layers, optimized with Adam and a learning rate of $0.001$.

**Datasets.** As is common in OCL studies, we evaluate on three image datasets: MNIST, Fashion-MNIST, and CIFAR-10. We deliberately avoid more complex datasets, since their substantially lower baseline accuracy would confound the analysis of errors attributable to label delays rather than model capacity. All benchmarks were evaluated under the Online Task-Incremental setting, where the dataset was divided into five disjoint tasks, each containing two classes.

**Evaluation.** We evaluated our experiments by running each experiment 30 times to mitigate chance findings, and we report the mean and variance across runs. For qualitative analysis, we plot per-task accuracy over time using Online Windowed Accuracy under a test-then-train protocol (computed per window). For quantitative analysis, we report final test-then-train Cumulative Accuracy, Accuracy Seen Avg, Accuracy All Avg, and Anytime Accuracy Avg. Cumulative Accuracy is defined as the average performance over all tasks evaluated at the end of training, making it a key metric to capture the trade-off between learning new information and retaining past knowledge. Accuracy Seen Avg measures the performance exclusively on previously seen tasks after completing training on each task. Accuracy All Avg measures the average accuracy across all tasks, including unseen ones, after the model completes training on each task. Anytime Accuracy Average quantifies a continuous learning model's performance throughout training. Computed by averaging periodic accuracy measurements taken during the learning with the results of Accuracy Seen and All. This metric captures both learning dynamics and knowledge retention, providing a more comprehensive view of model performance than final-task accuracy alone.

**Delay Setup.** In our experiments, we set the probability of no delay to 40%. If a batch is marked as delayed, it is presented to the system after 100 batches. These values were chosen to ensure that delayed batches are shown in both the current and the next task, while not extending beyond the duration of their originating task. We employed a buffer size of 128 instances for experience replay (ER) to balance memory efficiency with effective knowledge retention.

## 6 EXPERIMENTS

The first result (Figure 3) analyzes the impact of delays on the MNIST dataset. In all experiments, no delay was applied to the first task, simulating an initial deployment in which the system classifies specific targets whose data distribution changes over time.

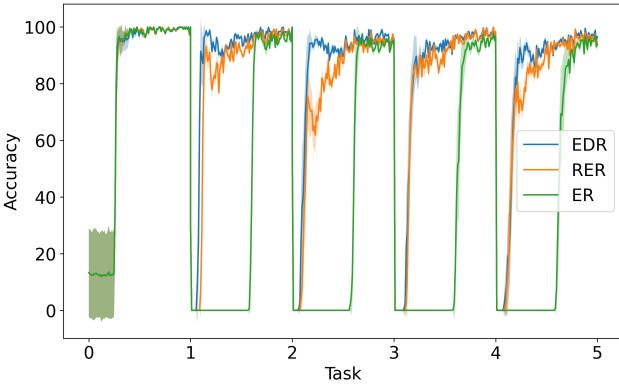

Figure 3: Accuracy on the MNIST dataset. After the initial task, with no delay, EDR recovers fastest, followed by RER, while ER shows the slowest recovery.

As expected, from the second task onward, ER requires more time to recover its previous performance. This occurs because ER was not originally designed to handle delayed data; hence, delayed instances from earlier tasks interfere with training on the current one. By randomly selecting instances (RER), instances from the current task are more likely to be sampled, which improves task performance. With EDR, the classifier recovers fastest by selecting instances that best support training, i.e., balancing loss and delay.

There are two important observations about the RER results. First, when labels (current and delayed) are uniformly distributed, this strategy tends to yield good performance; however, under skewed distributions with high delay levels, its behavior approaches that of ER. Second, for simple classification tasks, a small number of examples can restore prior performance, whereas more complex tasks require additional batches to recover.

In the following experiments (Figure 4), we compare the best OCL (EDR) method with a "fair" execution of ER. Here, ER performs multiple training passes to process all available instances, both delayed and current. Overall, the two approaches perform similarly, with EDR showing a slight advantage on some tasks. The main benefit, however, is efficiency: EDR reaches comparable (or better) accuracy with fewer training iterations.

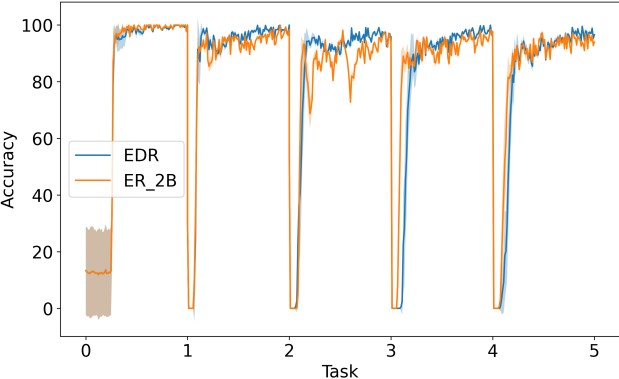

Figure 4: Comparing EDR and ER efficiency on MNIST. EDR and a robust version of ER (ER_2B) achieve similar peak accuracies. However, EDR reaches this performance with significantly fewer training iterations.

Using the same protocol, Figure 5 reports results on FashionMNIST. From the second task onward, both EDR and ER improve over the baseline, with EDR consistently leading. This gap aligns with EDR's prioritization criterion, balancing loss and delay, which better focuses updates on informative, timely instances as delayed data arrives.

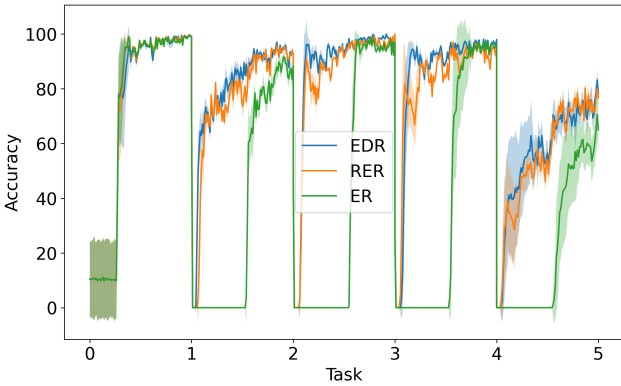

Figure 5: Accuracy on the FashionMNIST dataset. As previous results, EDR recovers fastest, followed by RER, while ER shows the slowest recovery.

Allowing ER to process all available examples (Figure 6), EDR and ER achieve similar accuracy. This parity is noteworthy for EDR: it reaches the same performance with fewer training updates (i.e., processing fewer examples), demonstrating superior sample/compute efficiency under delayed data.

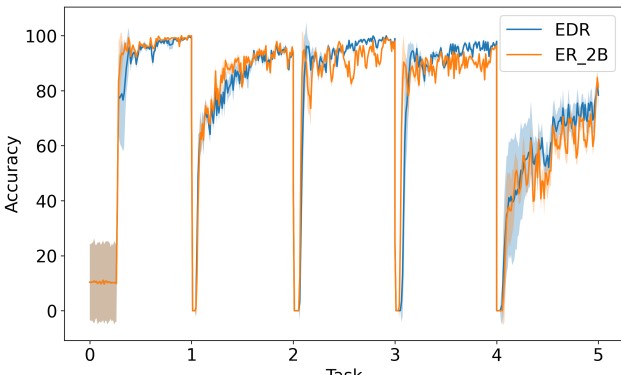

Figure 6: Comparing EDR and ER efficiency on FashionMNIST. EDR and a robust version of ER (ER_2B) achieve similar peak accuracies. Again, EDR reaches this performance with significantly fewer training iterations.

In the final experiment (Figure 7), we evaluate on the more complex CIFAR-10 dataset. Delays have a pronounced impact here: models require substantially more examples to recover prior performance. Across most settings, EDR achieves significantly better accuracy than the baselines. The exception is the extreme-delay condition, where all methods degrade because a portion of delayed instances (last task) falls beyond the evaluation window and never arrives.

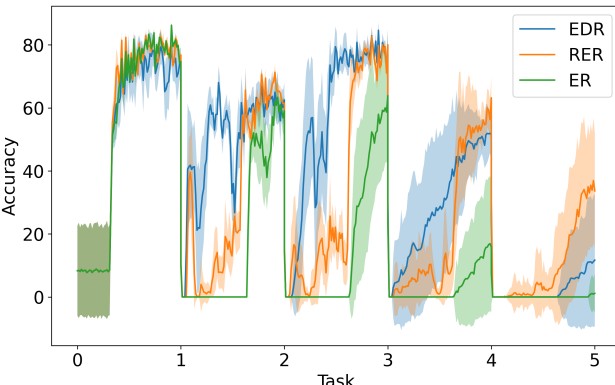

Figure 7: Accuracy on the CIFAR-10 dataset. In a more complex dataset, EDR achieves significantly better accuracy than RER and ER.

Finally, Figure 8 shows that EDR delivers even stronger results. Unlike ER, which requires repeated passes to process all available instances, EDR prioritizes the most relevant data (i.e., low-loss, current-task examples) and limits the influence of delayed labels during updates, yielding consistent gains in overall accuracy with fewer training iterations.

After performing a visual inspection, we have analyzed the performance of the OCL methods in a quantitative way. Table 1 mirrors the visual trends: across MNIST and FashionMNIST, EDR is best or co-best on all aggregate metrics, matching (MNIST) or slightly exceeding (FashionMNIST) the "fair" ER variant (ER_2B) in *Cumulative Accuracy* while maintaining strong *Seen*, *All*, and *Anytime* averages. RER is competitive in streaming-style performance and attains the highest *Anytime* accuracy on FashionMNIST, consistent with uniform replay sampling more current-task examples; however, its end-of-stream accuracy remains below EDR. On the more challenging CIFAR-10 set-

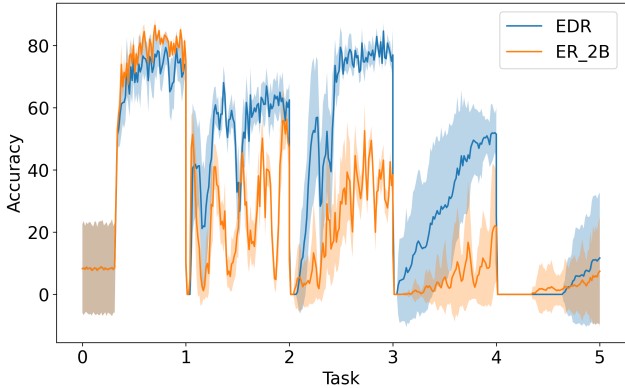

Figure 8: Comparing EDR and ER efficiency on CIFAR-10. In this experiment, using a more complex dataset, EDR presented the best overall results.

ting, delays have a larger impact and EDR separates clearly on every metric (e.g., markedly higher *Cumulative Accuracy*) compared to both RER and ER_2B, indicating that prioritizing informative, timely instances, by balancing loss and delay, becomes increasingly important as task complexity and delay effects grow.

Table 1: Comparison of Cumulative Accuracy, Accuracy on Seen tasks, Accuracy on All tasks, and Anytime Accuracy across different strategies (EDR, RER, ER, ER-2B) on MNIST, Fashion-MNIST, and CIFAR-10 datasets.

| Dataset | Strategy | Cum. Acc. | Acc. Seen Avg. | Acc. All Avg. | Anytime Acc. Avg. |
|---|---|---|---|---|---|
| MNIST | EDR | $81.66 \pm 1.11$ | $34.98 \pm 0.11$ | $21.00 \pm 0.06$ | $20.35 \pm 0.09$ |
| | RER | $78.12 \pm 1.22$ | $34.74 \pm 0.15$ | $20.81 \pm 0.08$ | $20.15 \pm 0.08$ |
| | ER | $45.01 \pm 1.05$ | $33.96 \pm 0.17$ | $20.33 \pm 0.10$ | $18.66 \pm 0.10$ |
| | ER_2B | $81.83 \pm 1.06$ | $35.84 \pm 0.16$ | $21.50 \pm 0.08$ | $20.44 \pm 0.08$ |
| Fashion MNIST | EDR | $75.87 \pm 1.77$ | $34.18 \pm 0.33$ | $20.70 \pm 0.17$ | $19.93 \pm 0.13$ |
| | RER | $73.65 \pm 1.17$ | $33.98 \pm 0.21$ | $20.69 \pm 0.11$ | $20.69 \pm 0.11$ |
| | ER | $42.42 \pm 1.55$ | $32.63 \pm 0.37$ | $19.86 \pm 0.31$ | $18.35 \pm 0.24$ |
| | ER_2B | $75.12 \pm 1.25$ | $34.42 \pm 0.18$ | $20.73 \pm 0.13$ | $20.14 \pm 0.10$ |
| CIFAR10 | EDR | $37.59 \pm 3.00$ | $19.11 \pm 0.74$ | $11.03 \pm 0.41$ | $10.92 \pm 0.35$ |
| | RER | $30.66 \pm 3.05$ | $20.31 \pm 0.42$ | $11.99 \pm 0.32$ | $11.54 \pm 0.27$ |
| | ER | $18.04 \pm 2.30$ | $18.43 \pm 0.60$ | $10.65 \pm 0.22$ | $10.69 \pm 0.24$ |
| | ER_2B | $22.06 \pm 3.02$ | $19.25 \pm 0.38$ | $11.07 \pm 0.30$ | $11.08 \pm 0.26$ |

## 7 FINAL REMARKS

In this work, we introduced a novel investigation on online continual learning in the realistic setting of delayed labels. Our empirical analysis demonstrates that label latency introduces a critical challenge by entangling training signals from past and current tasks, leading to degraded performance. To address this, we proposed Experience-Delayed Replay (EDR), a delay-aware instance selection strategy that intelligently prioritizes which late-labeled examples to use for model updates. Our results show that EDR consistently outperforms delay-agnostic baselines, including ER and its random variant (RER), by accelerating performance recovery after task shifts and improving current-task accuracy and stability. By selectively using only the most beneficial delayed instances, EDR achieves these improvements while using a smaller training budget, highlighting a clear efficiency advantage. This work not only provides a foundational understanding of the delayed-label problem but also offers a practical solution compatible with existing replay-based CL methods, paving the way for more robust and realistic continual learning systems. Limitations include our focus on supervised classification and assumed access to label arrival times, future work includes extending EDR to unsupervised or self-supervised streams, and adaptive delay estimation.

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
