# OpenReview forum: "Look Back to Move Forward: Delay-Aware Instance Selection for Online Continual Learning"
_ICLR.cc/2026/Conference — ICLR 2026 Conference Withdrawn Submission_

### Official Review · Reviewer_Ry5o · 2025-10-30

**Soundness:** 2
**Presentation:** 2
**Contribution:** 1
**Rating:** 2
**Confidence:** 5

**Summary:**

The proposed approach focuses on a challenging variant of continual learning research, continual learning with label delay, which is mostly ignored by the existing baselines. Most notable baselines in CL research mainly assume that each incremental batches arrive with its corresponding labels; however, in a real-world setup, that may or may not hold. Therefore, this problem setting is challenging and highly relevant in this domain. The authors proposed a variant of experience replay to adapt to such conditions, with empirical results validating the effectiveness of the proposed approach.

**Strengths:**

1. Label delay in CL setup is a relevant challenge, yet most of the existing baselines assume the availability of labels during each incremental learning. This paper aims to address that issue.

2. This paper proposes a new approach addressing the label delay, EDR, which can be easily integrated into existing ER based CL approaches.

3. Empirical results demonstrate the effectiveness of the proposed approach over experienced replay in CL.

**Weaknesses:**

1. The core contribution of this paper is limited to Equation 1, which is somewhat incremental with respect to experience replay as a continual learning approach. This paper does not provide any theoretical justification behind the design of such an approach.

2. This paper does not provide any ablation study highlighting the various aspects of the hyperparameters that could affect the model's performance.

3. This paper only compares against ER as a baseline; however, multiple CL baselines exist that could be modified or adapted to this setup. These comparisons could demonstrate why the proposed approach is needed over existing ones. Furthermore, this paper only evaluates on variants of MNIST and CIFAR-10 datasets, which is very limiting. It would be beneficial if the authors compared performance on a large-scale dataset such as ImageNet-1K.

**Questions:**

Please refer to the weaknesses section.

---

### Official Review · Reviewer_Bwf1 · 2025-10-31

**Soundness:** 2
**Presentation:** 2
**Contribution:** 2
**Rating:** 2
**Confidence:** 5

**Summary:**

This paper introduces Experience-Delayed Replay (EDR), a simple, delay-aware instance selection method for online continual learning (OCL) when labels arrive late and formalize the setting and analyze how varying delay affect learning dynamics, and propose an instance-selection mechanism that prioritizes delayed examples based on their loss or confidence and delay.

**Strengths:**

1. Clearly defines and formalizes label delay in online continual learning which is a realistic and underexplored setting.
2. EDR is a minimal modification to standard ER, making it easy to implement and compatible with existing pipelines.
3. Addresses annotation latency, a real concern in streaming systems.

**Weaknesses:**

1. Limited novelty at the algorithmic level: EDR’s main innovation is a weighted replay criterion which is conceptually simple and incremental.
2. Only uses toy/moderate datasets (MNIST, FashionMNIST, CIFAR-10) with shallow MLPs; unclear scalability to deeper architectures or complex streams with long-tailed, imbalanced and blurry setups to strengthen the weighted replay proposed.
3. Results focus mainly on accuracy metrics; lacks ablations on sensitivity to delay hyperparameters (k, weighting form) or computational overhead and also critical metrics such as forgetting, forward and backward transfer which would give more insights on how delayed labels benefit earlier class learning.
4. The method presumes access to delay magnitudes (∆t), which might not be available in practice.

**Questions:**

1. Would EDR’s benefits persist for larger models or datasets (e.g., ImageNet, ViT backbones)? Any insights from preliminary runs?
2. How does EDR interact with regularization-based or architectural continual learning methods?
3. Do you have further insights in terms of catastrophic forgetting, forward and backward transfer metrics?
4. How realistic is the assumed 100-batch delay? Does EDR generalize to stochastic or task-dependent delay distributions?

---

### Official Review · Reviewer_iakK · 2025-11-01

**Soundness:** 1
**Presentation:** 2
**Contribution:** 1
**Rating:** 2
**Confidence:** 4

**Summary:**

This paper primarily focuses on the label delay problem in online continual learning (OCL). For memory-based methods, while the vanilla Experience Replay (ER) suffers from label delay, the author examines how such delays impact the training of ER under various latency scenarios. Additionally, a memory-based method called Experience Delay Repair (EDR) is proposed to mitigate the performance loss associated with label delay.

**Strengths:**

1. The proposed method is clear and intuitive.
2. There are some advantages of the proposed method compared with the standard ER.
3. The source code is included in the submission.

**Weaknesses:**

### Major Concerns

1. **Unclear Data Source for Figure 1**: It is not clear what the data source for Fig. 1 is. Is it based on experimental results, or is it a conceptual diagram? If it represents results, I do not observe a significant difference in terms of final accuracy. If it is a conceptual diagram, what experimental evidence supports such claims? Additionally, the point at which the task shift occurs in this figure is not indicated.

2. **Lack of Empirical Evidence**: There is insufficient proof for the statement, “Our empirical study shows that the nature of the delay regime matters. With small delays, supervision drift is mild but non-negligible; with moderate delays, the training signal substantially overlaps multiple tasks; with large or bursty delays, the learner faces update floods where labels from several past tasks arrive simultaneously. In all cases, blindly consuming every delayed label wastes updates and can bias the model away from the current distribution.” Please include the results from the empirical study to substantiate this claim.

3. **Questionable Contribution**: Contribution 1 claimed in line 22 is not a legitimate contribution, as the definition of label delay has been proposed in prior research [A] from NeurIPS 2024. Furthermore, there is no comparison made with this work in the manuscript, which could be detrimental to this submission.

4. **Accuracy Claims about EDR**: The assertion that “EDR achieves significantly better accuracy than RER and ER” is misleading, according to Fig. 7, where RER exhibited the best performance at the end of training.

5. **Scalability of Proposed Method**: The scalability of the proposed method is not adequately addressed. Traditionally, OCL is evaluated using ResNet-18 or Vision Transformers, as this is a standard protocol. In prior research concerning label delay in OCL, ResNet-18 was also used in the main experiments. This manuscript employs a Multi-Layer Perceptron (MLP), raising questions about whether the experimental findings will generalize to conventional models like ResNet-18.

6. **Evaluation of "Plug-and-Play" Method**: While the author claims that the proposed method is “plug-and-play” in line 71, only ER is evaluated. To substantiate this claim, it is suggested that results from other state-of-the-art memory-based methods, such as OCM [B], GSA [C], MOSE [D], and CCLDC [E], be included.

### Minor Concerns

- There is a missing reference in line 158.

**References:**
- [A] Csaba, Botos, et al. "Label delay in online continual learning." Advances in Neural Information Processing Systems 37 (2024): 119976-120012.
- [B] Guo, Yiduo, Bing Liu, and Dongyan Zhao. "Online continual learning through mutual information maximization." International Conference on Machine Learning. PMLR, 2022.
- [C] Guo, Yiduo, Bing Liu, and Dongyan Zhao. "Dealing with cross-task class discrimination in online continual learning." Proceedings of the IEEE/CVF Conference on Computer Vision and Pattern Recognition. 2023.
- [D] Yan, Hongwei, et al. "Orchestrate latent expertise: Advancing online continual learning with multi-level supervision and reverse self-distillation." Proceedings of the IEEE/CVF Conference on Computer Vision and Pattern Recognition. 2024.
- [E] Wang, Maorong, et al. "Improving plasticity in online continual learning via collaborative learning." Proceedings of the IEEE/CVF Conference on Computer Vision and Pattern Recognition. 2024.

**Questions:**

See above

---

### Note · Authors · 2025-11-19

I have read and agree with the venue's withdrawal policy on behalf of myself and my co-authors.